# Hierarchically Integrated Models: Learning to Navigate from Heterogeneous Robots

**Katie Kang**
UC Berkeley

**Gregory Kahn**
UC Berkeley

**Sergey Levine**
UC Berkeley

**Abstract:** Deep reinforcement learning algorithms require large and diverse datasets in order to learn successful policies for perception-based mobile navigation. However, gathering such datasets with a single robot can be prohibitively expensive. Collecting data with multiple different robotic platforms with possibly different dynamics is a more scalable approach to large-scale data collection. But how can deep reinforcement learning algorithms leverage such heterogeneous datasets? In this work, we propose a deep reinforcement learning algorithm with hierarchically integrated models (HInt). At training time, HInt learns separate perception and dynamics models, and at test time, HInt integrates the two models in a hierarchical manner and plans actions with the integrated model. This method of planning with hierarchically integrated models allows the algorithm to train on datasets gathered by a variety of different platforms, while respecting the physical capabilities of the deployment robot at test time. Our mobile navigation experiments show that HInt outperforms conventional hierarchical policies and single-source approaches.

**Keywords:** Deep reinforcement learning, Multi-robot learning, Autonomous navigation

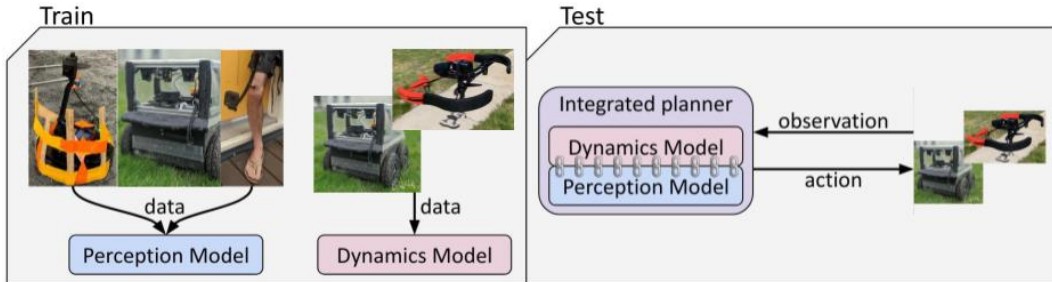

Figure 1: Overview of our hierarchically integrated models (HInt) algorithm. At training time, HInt separately trains a perception model and a dynamics model, then at test time, HInt combines the perception and dynamics model into a single model for integrated planning and execution. Our modular training procedure enables HInt to train the perception model using data gathered by multiple platforms, such as ground robots and even people recording video with a hand-held camera, while our integrated model at test time ensures the perception model only considers trajectories which are dynamically feasible.

## 1 Introduction

Machine learning provides a powerful tool for enabling robots to acquire control policies by learning directly from experience. One of the key guiding principles behind recent advances in machine learning is to leverage large datasets. In previous works in deep reinforcement learning for robotic control, the most common approach is to collect data from a single robot, train a policy in an end-to-end fashion, and deploy the policy on the same data-collection platform. This approach necessitates collecting a large and diverse dataset for every model of robot we wish to deploy on, which can present a significant practical obstacle, since there exists a plethora of different platforms in the real world. What if we could instead leverage datasets collected by a variety of *different* robots in our training procedure? An ideal method could use data from *any* platform that provides useful knowledge about the problem. As an example, it is expensive and time-consuming to gather a large dataset for visual navigation on a drone, due to on-board battery constraints. In comparison, it is

5th Conference on Robot Learning (CoRL 2021), London, UK.

much easier to collect data on a robot car. Because knowledge about the visual features of obstacles can be shared across vision-based mobile robots, making use of visual data collected by a car to train a control policy for a drone could significantly reduce the amount of data needed from the drone. Unfortunately, data from such heterogeneous platforms presents a challenge: different platforms have different physical capabilities and action spaces. In order to leverage such heterogeneous data, we must properly account for the underlying dynamics of the data collection platform.

To learn from multiple sources of data, previous works have utilized hierarchical policies [1, 2, 3]. In this type of method, a high-level and a low-level policy are trained separately. At test time, the actions generated by the high-level policy are used as waypoints for the low-level policy. By separating the policy into two parts, these algorithm are able to utilize data from multiple sources in the high-level policy, while representing robot-specific information such as the dynamics in the low-level policy. One drawback of these methods, however, is that the low-level policy is unable to communicate any robot-specific information to the high-level policy. This makes it possible for the high-level policy to command waypoints that are impossible for the robot to physically achieve, leading to poor performance and possibly dangerous outcomes.

The key idea in our approach, illustrated in Fig. 1, is to instead learn hierarchical *models*, and to integrate the hierarchical models for planning. At training time, we learn a perception model that reasons about interactions with the world, using our entire multi-robot dataset, and a dynamics model specific to the deployment robot, using only data from that robot. At test time, the perception and dynamics models are combined to form a single integrated model, which a planner uses to choose the actions. Such hierarchically integrated models can leverage data from multiple robots for the perception layer, while also reasoning about the physical capabilities of the deployment robot during planning. This is because when the algorithm uses the hierarchically integrated model to plan, the dynamics model can "hint" to the perception model about the robot-specific physical capabilities of the deployment robot, and the planner can only select behaviors that the deployment robot can actually execute (according to the dynamics model). In contrast to hierarchical policies, which may produce waypoint commands that are dynamically infeasible for the deployment robot to achieve, our hierarchically integrated models take the robot's physical capabilities into account, and only permit physically feasible plans.

The primary contribution of this work is HInt— hierarchically integrated models for acquiring image-based control policies from heterogeneous datasets. We demonstrate that HInt successfully learns policies from heterogeneous datasets in real-world navigation tasks, and outperforms methods that use only one data source or use conventional hierarchical policies.

## 2   Related Work

Prior work demonstrated end-to-end learning for vision-based control for a wide variety of applications, from video games [4, 5] and manipulation [6, 7] to navigation [8, 9]. However, these approaches typically require a large amount of diverse data [10], which hinders the adoption of these algorithms for real-world robot learning. One approach for overcoming these data constraints is to combine data from multiple robots; While prior methods have addressed collective learning, they typically assume that the robots are the same [11], have similar underlying dynamics [12], or the data is from expert demonstrations [13, 14, 15]. Our approach learns from off-policy data gathered by robots with a variety of underlying dynamics. Prior methods have also transferred skills from simulation, where data is more plentiful [16, 17, 18, 19, 20, 21, 22]. In contrast, our method does not require any simulation, and instead is aimed at leveraging heterogeneous real-world data sources, though it could be extended to incorporate simulated data as well.

Prior work has also investigated learning hierarchical vision-based control policies for applications such as autonomous flight [23, 2] and driving [1, 24, 3, 25]. One advantage of these conventional hierarchy approaches is that many can leverage heterogeneous datasets [26, 12, 23]. However, *even if each module is perfectly accurate*, these conventional hierarchy approaches can still fail because the low-level policy cannot communicate the robot's capabilities and limitations to the high-level policy. In contrast, because HInt learns hierarchical models, and performs planning on the integrated models at test time, HInt is able to jointly reason about the capabilities of the robot and the perceptual features of the environment.

End-to-end algorithms that can leverage datasets from heterogeneous platforms have also been investigated by prior work; however, these methods typically require on-policy data, are evaluated in

visually simplistic domains, or have only been demonstrated in simulation [27, 28, 29]. In contrast, HInt works with real-world off-policy datasets because at the core of HInt are predictive models, which can be trained using standard supervised learning.

## 3 HInt: Hierarchically Integrated Models

Our goal is to learn image-based mobile navigation policies that can leverage data from heterogeneous robotic platforms. The key contribution of our approach, shown in Fig. 2, is to learn separate hierarchical models at training time, and combine these models into a single integrated model for planning at test time. The two hierarchical models include a high-level, shared perception model, and a low-level, robot-specific dynamics model. The perception model can be trained using data gathered by a variety of different robots, all with possibly different dynamics, while the robot-specific dynamics model is trained only using data from the deployment robot. At test time, because the output predictions of the dynamics model —which are kinematic poses —are the input actions for the perception model, the dynamics and perception models can be combined into a single integrated model. In the following sections, we will describe how HInt trains a perception model and a dynamics model, combines these models into a single integrated model in order to perform planning, and conclude with an algorithm summary.

### 3.1 Perception Model

The perception model is a neural network predictive model $f_\theta^{\mathrm{PER}}(\mathbf{o}_t, \boldsymbol{\delta}\mathbf{p}_{t:t+H}) \to \hat{r}_{t:t+H}$—shown in Fig. 2— parameterized by model parameters $\theta$. $f_\theta^{\mathrm{PER}}$ takes as input the current image observation $\mathbf{o}_t$ and a sequence of $H$ future changes in poses $\boldsymbol{\delta}\mathbf{p}_{t:t+H} = (\boldsymbol{\delta}\mathbf{p}_t, \boldsymbol{\delta}\mathbf{p}_{t+1}, ..., \boldsymbol{\delta}\mathbf{p}_{t+H-1})$, and predicts $H$ future rewards $\hat{r}_{t:t+H} = (\hat{r}_t, \hat{r}_{t+1}, ..., \hat{r}_{t+H-1})$. The pose $\mathbf{p}$ characterizes the kinematic configuration of the robot. In our experiments, we used the robot's x position, y position, and yaw orientation as the pose, though 3D configurations that include height, roll, and pitch are also possible. This kinematic configuration provides a dynamics-agnostic interface between the perception model and the dynamics model. The perception model is trained using the heterogeneous dataset $\mathcal{D}^{\mathrm{PER}}$ by minimizing the objective:

$$\mathcal{L}^{\mathrm{PER}}(\theta, \mathcal{D}^{\mathrm{PER}}) = \sum_{(\mathbf{o}_t, \boldsymbol{\delta}\mathbf{p}_{t:t+H}, r_{t:t+H})} \|\hat{r}_{t:t+H} - r_{t:t+H}\|_2^2 \tag{1}$$
$$\hat{r}_{t:t+H} = f_\theta^{\mathrm{PER}}(\mathbf{o}_t, \boldsymbol{\delta}\mathbf{p}_{t:t+H})$$

The perception model can be trained with a heterogeneous dataset consisting of data gathered by a variety of robots, all with possibly different underlying dynamics. The only requirement for the data is that the recorded sensors (e.g., camera) are approximately the same, and that we can calculate the change in pose—position and orientation—between each sequential datapoint, which could be done using visual or mechanical odometry, as well as the reward, which could be collected via onboard sensors or manual labeling. This ability to train using datasets gathered by heterogeneous platforms is crucial because the perception model is an image-based neural network, which requires large amounts of data in order to effectively generalize.

### 3.2 Dynamics Models

The dynamics model is a neural network predictive model $f_\phi^{\mathrm{DYN}}(\mathbf{s}_t, \mathbf{a}_{t:t+H}) \to \hat{\boldsymbol{\delta}\mathbf{p}}_{t:t+H}$—shown in Fig. 2— parameterized by model parameters $\phi$. $f_\phi^{\mathrm{DYN}}$ takes as input the current robot state $\mathbf{s}_t$ and

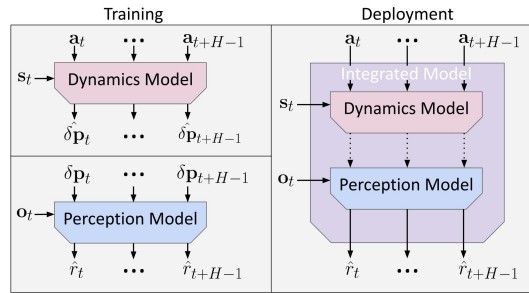

Figure 2: During training, we learn two separate neural network models. The dynamics model takes as input the current robot state and a sequence of future actions, and predicts future changes in poses. This model is trained using data gathered by a single robot. The perception model takes as input the current image observation and a sequence of future changes in poses, and predicts future rewards. This model is trained using data gathered by a variety of robots that have the same image observations, but potentially different dynamics. When deploying, the dynamics and perception models are combined into a single integrated model that is used for planning and executing actions that maximize reward.

a sequence of $H$ future actions $\mathbf{a}_{t:t+H} = (\mathbf{a}_t, \mathbf{a}_{t+1}, ..., \mathbf{a}_{t+H-1})$, and predicts $H$ future changes in poses $\hat{\boldsymbol{\delta}\mathbf{p}}_{t:t+H} = (\hat{\boldsymbol{\delta}\mathbf{p}}_t, \hat{\boldsymbol{\delta}\mathbf{p}}_{t+1}, ..., \hat{\boldsymbol{\delta}\mathbf{p}}_{t+H-1})$. The state $\mathbf{s}$ can include any sensor information, such as the robot's battery charge or velocity, that may help the model to make more accurate predictions of future poses. The dynamics model is trained using the respective robot-specific dataset $\mathcal{D}^{\text{DYN}}$ by minimizing the objective:

$$\mathcal{L}^{\text{DYN}}(\phi, \mathcal{D}^{\text{DYN}}) = \sum_{(\mathbf{s}_t, \mathbf{a}_{t:t+H}, \boldsymbol{\delta}\mathbf{p}_{t:t+H})} \|\hat{\boldsymbol{\delta}\mathbf{p}}_{t:t+H} - \boldsymbol{\delta}\mathbf{p}_{t:t+H}\|_2^2 \qquad (2)$$

$$\hat{\boldsymbol{\delta}\mathbf{p}}_{t:t+H} = f_\phi^{\text{DYN}}(\mathbf{s}_t, \mathbf{a}_{t:t+H}).$$

Although the dynamics model must only be trained using data collected by the deployment robot, the dynamics model dataset can be significantly smaller compared to the perception model dataset, because the dynamics model inputs are lower dimensional by orders of magnitude. Furthermore, the dynamics model dataset does not need a reward signal, allowing for easier data collection. Though, because perception and dynamics are decoupled in HInt, the dynamics model can accurately model systems whose dynamics depend on low-dimensional state information only.

### 3.3 Planning and Control

In order to perform planning and control at test time, we first combine the perception and dynamics models into a single integrated model $f_{\theta,\phi}^{\text{COM}}(\mathbf{o}_t, \mathbf{s}_t, \mathbf{a}_{t:t+H}) = f_\theta^{\text{PER}}(\mathbf{o}_t, f_\phi^{\text{DYN}}(\mathbf{s}_t, \mathbf{a}_{t:t+H})) \rightarrow \hat{r}_{t:t+H}$, shown in Fig. 2. These models can be combined without any additional training because the output of the dynamics model—changes in kinematic poses—is also the input to the perception model. This integrated model is essential because it enables the planner to holistically reason about the entire system. In contrast, in conventional hierarchical control methods, where the high-level policy outputs a goal for the low-level policy, the high-level policy could output a dynamically infeasible reference trajectory for the low-level controller. Our approach would not suffer from this failure case because, with integrated planning, the perception model can only take as input dynamically-feasible trajectories that are output by the dynamics model.

We then plan at each time step for the action sequence that maximizes reward according to the integrated model by solving the following optimization:

$$\mathbf{a}_{t:t+H}^* = \arg \max_{\mathbf{a}_{t:t+H}} R(\hat{r}_{t:t+H}, \hat{\boldsymbol{\delta}\mathbf{p}}_{t:t+H}, \mathbf{a}_{t:t+H}, \mathbf{g}_t) \qquad (3)$$

$$\hat{r}_{t:t+H} = f_{\theta,\phi}^{\text{COM}}(\mathbf{o}_t, \mathbf{s}_t, \mathbf{a}_{t:t+H}).$$

Here, $R$ is a user-defined task-specific reward function, and $\mathbf{g}_t$ is a user-specified goal. We follow the framework of model predictive control, where at each time step, the robot calculates the best sequence of actions for the next $H$ steps, executes the first action in the sequence, and then repeats the process at the next time step. The action sequence that approximately maximizes the objective in Eqn. 3 can be computed using any optimization method. In our implementation, we employ stochastic zeroth-order optimization, which selects the best sequence among a set of randomly generated trajectories, as is common in model-based RL [30, 31]. Specifically, we used either the cross-entropy method (CEM) [32] or MPPI [33], depending on the computational constraints of the platform. While this approach does result in a limitation for long-horizon reasoning, our main focus is to perform short-horizon navigation.

### 3.4 Algorithm Summary

We now briefly summarize how our HInt system operates during training and deployment, as shown in Fig. 1 and Fig. 2.

During training, we first gather perception data using a number of platforms. For each of these platforms, we save the onboard observations $\mathbf{o}$, change in poses $\boldsymbol{\delta}\mathbf{p}$, and rewards $r$ into a shared dataset $\mathcal{D}^{\text{PER}}$ (see §B.1), and use this dataset to train the perception model $f_\theta^{\text{PER}}$ (Eqn. 1) (see §B.2). Then, using the robot we will deploy at test time, we gather dynamics data by having the robot act in the real world and recording the robot's onboard states $\mathbf{s}$, actions $\mathbf{a}$, and change in poses $\boldsymbol{\delta}\mathbf{p}$ into the dataset $\mathcal{D}^{\text{DYN}}$ (see §C.1); we use this dataset to train the dynamics model $f_\phi^{\text{DYN}}$ (Eqn. 2) (see §C.2).

When deploying HInt, we first combine the perception model $f_\theta^{\text{PER}}$ and dynamics model $f_\phi^{\text{DYN}}$ into a single model $f_{\theta,\phi}^{\text{COM}}$. The robot then plans a sequence of actions that maximizes reward (Eqn. 3), executes the first action, and repeat this planning process at each time step until the task is complete, as is standard in model-based RL with model-predictive control [30, 31] (see §D).

| Indoor Robot in Office | Outdoor Robot in Urban | Person in Industrial | Drone in Urban |
|---|---|---|---|

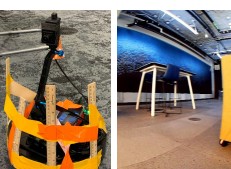 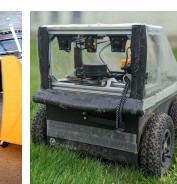 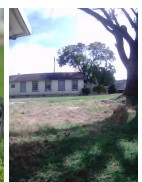 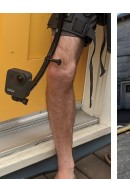 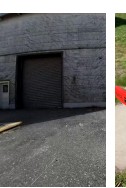 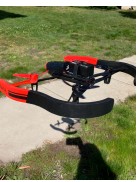 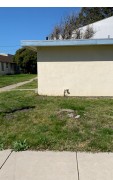

Figure 3: Training data was gathered by an indoor Yujin Kobuki robot in an office, an outdoor Clearpath Jackal robot in an urban environment, and a person with a video camera in an industrial area. HInt is able to jointly train on this heterogeneous data, which leads to improved navigation performance. HInt was deployed on the ourdoor Jackal robot as well as a Parrot Bebop drone in the urban environment. Because HInt is able to reason about the dynamics of the deployment robot during planning, it is able to successfully control robots that are not included in the perception dataset, such as the drone.

## 4  Experiments

We now present an experimental evaluation of our hierarchically integrated models algorithm in the context of real world mobile robot navigation tasks. Videos and additional details about the data sources, models, training procedures, and planning can be found in the Appendix section or on the project website: https://sites.google.com/berkeley.edu/hint-public

In our experiments, we aim to answer the following questions:

**Q1**: Does leveraging data collected by other robots, in addition to data from the deployment robot, improve performance compared to only learning from data collected by the deployment robot?

**Q2**: Does HInt's integrated model planning approach result in better performance compared to conventional hierarchy approaches?

In order to separately examine these questions, we investigate **Q1** by training the perception module with multiple different real-world data sources, including data from different environments and different platforms, and evaluating on a single real-world robot. To examine **Q2**, we deploy a shared perception module to systems with different low-level dynamics.

### 4.1  Q1: Comparison to Single-Source Models

In this experiment, we deployed a robot in a number of visually diverse environments, including ones in which *that robot itself has not collected any data*. Our hypothesis is that HInt, which can make use of heterogeneous datasets gathered by multiple robots, will outperform methods that can only train using data gathered by the deployment robot.

Perception data was collected in three different environments using three different platforms (Fig. 3): a Yujin Kobuki robot in an indoor **office** environment (3.7 hours), a Clearpath Jackal in an outdoor **urban** environment (3.5 hours), and a person recording video with a hand-held camera in an **industrial** environment (1.2 hours). The deployment robot is the Clearpath Jackal. The same dataset that provided the Jackal perception data as described above was also used for the Jackal dynamics data.

The robot's objective is to drive towards a goal location while avoiding collisions and minimizing action magnitudes. More specifically, the reward $r$ is $-1$ for a collision and $0$ otherwise, the goal $\mathbf{g}$ specifies a GPS location, $\angle(\hat{\boldsymbol{\delta}}\mathbf{p}_{t+h}, \mathbf{g}_t)$ denotes the angle in radians formed by $\hat{\boldsymbol{\delta}}\mathbf{p}_{t+h}$ and $\mathbf{g}_t$ with respect to the robot's current position, and the reward function used for planning is

$$R(\hat{r}_{t:t+H}, \hat{\boldsymbol{\delta}}\mathbf{p}_{t:t+H}, \mathbf{a}_{t:t+H}, \mathbf{g}_t) \tag{4}$$
$$= \sum_{h=0}^{H-1} \hat{r}_{t+h} - 0.3 \cdot \angle(\hat{\boldsymbol{\delta}}\mathbf{p}_{t+h}, \mathbf{g}_t) - 0.05 \cdot \|\hat{\boldsymbol{\delta}}\mathbf{p}_{t+h}\|_1.$$

We compared HInt, which learns from all the data sources, with a variant of HInt which only learned from a single data source. Both approaches use the same training parameters and neural network architecture for the integrated model in order to make the most fair comparison, with the only difference between the data with which they were trained. In our implementation, HInt and the single data source approach both build upon the method proposed in Kahn et al [34]. Kahn et al [34] also uses a vision-based predictive model to perform planning, but differs from HInt in the following ways: (i) only data gathered by the deployment robot can be used for training and (ii) the integrated perception and dynamics model is trained end-to-end.

| | | Perception Data Sources | | | | |
|---|---|---|---|---|---|---|
| | | Single-Source | | | HInt (ours) | |
| | | Kobuki (Office) | Jackal (Urban) | Human (Industrial) | Kobuki (Office) + Jackal (Urban) | Kobuki (Office) + Jackal (Urban) + Human (Industrial) |
| Test Env | Urban | 0% | 13% | N/A | **100%** | **100%** |
| | Industrial | 0% | N/A | 0% | 33% | **87%** |

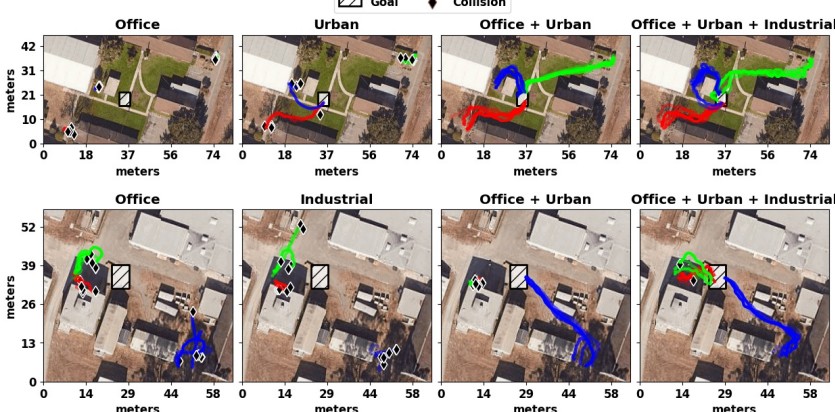

Figure 4: Comparison of single data source models versus our HInt approach in an urban and industrial environment for the task of reaching a goal location while avoiding collisions using the Clearpath Jackal robot. Note that HInt was trained with more datapoints than the single source approach, because HInt is able to learn from data collected by other platforms in addition to the Jackal deployment robot. Each approach was evaluated from the 3 same start locations in each environment (corresponding to the red, green, and blue lines), and was ran 5 times from each start location. The images show a top down view of the Urban (top row) and Industrial (bottom row) environments, with each column corresponding to the data sources used to train the perception model. The quantitative results show what percentage of the 15 trials successfully reached the goal.

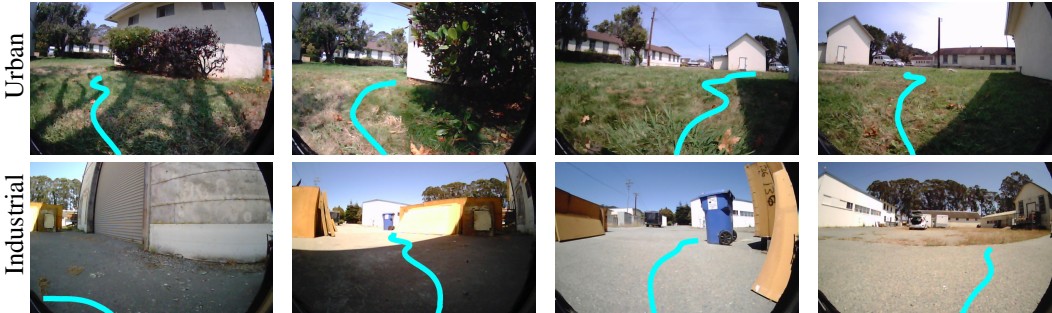

Figure 5: Visualization of HInt at test time successfully reaching the goal location while avoiding collisions in urban (top) and industrial (bottom) environments.

Fig. 4 shows the results comparing HInt to the single data source approach. In all environments[1], our approach is more successful in reaching the goal. Note that even when the single data source method is trained and deployed in the same environment, HInt still performs better, because learning-based methods benefit from large and diverse datasets. Furthermore, the row labeled "Industrial" illustrates well how HInt can benefit from data collected with other platforms: although the Jackal robot had never seen the industrial setting during training, the training set did include data collected by a person with a video camera in this setting. The increase in performance from including this data ("Kobuki + Jackal + Human") shows that the Jackal robot was able to effectively integrate this data into its visual navigation strategy. Fig. 5 shows first-person images of HInt successfully navigating in both the urban and industrial environments.

## 4.2 Q2: Comparison to Conventional Hierarchy
In this experiment, we deployed the perception module onto robots with different low-level dynamics, including ones that were not seen during the collection of the perception data. Our hypothesis is

---

[1]We could not run experiments in the office environment due to COVID-related closures.

that our integrated HInt approach will outperform conventional hierarchy approaches, because it is able to jointly reason about perception and dynamics.

We compared our integrated approach with the most commonly used hierarchical approach, in which the perception model is used to output desired waypoints that are then passed to a low-level controller [1, 23, 24, 2, 3]. In our prediction- and planning-based framework, this modular approach is implemented by first planning over a sequence of positions $\boldsymbol{\delta p}^*_{t:t+H}$ by using a kinematic vision-based model, and then planning over a sequence of actions $\mathbf{a}^*_{t:t+H}$ so as to minimize the tracking error against these planned positions using a robot-specific dynamics model. This baseline represents a clean "apples-to-apples" comparison between our approach – which directly combines both the dynamic and kinematic models into a single model – and a conventional pipelined approach that separates vision-based kinematic planning with low-level trajectory tracking. Both our method and this baseline employ the same neural network architectures for the vision and dynamics models, and train them on the same data, thus providing for a controlled comparison that isolates the question of whether our end-to-end approach improves over a conventional pipelined hierarchical approach. The planning process for the hierarchical baseline can be expressed as the following two-step optimization:

$$\boldsymbol{\delta p}^*_{t:t+H} = \arg \max_{\boldsymbol{\delta p}_{t:t+H}} R(\hat{r}_{t:t+H}, \boldsymbol{\delta p}_{t:t+H}, \mathbf{g}_t) \tag{5}$$

$$\hat{r}_{t:t+H} = f^{\text{PER}}_\theta(\mathbf{o}_t, \boldsymbol{\delta p}_{t:t+H})$$

$$\mathbf{a}^*_{t:t+H} = \arg \min_{\mathbf{a}_{t:t+H}} \sum_{h=0}^{H-1} \|\hat{\boldsymbol{\delta p}}_{t+h} - \boldsymbol{\delta p}^*_{t+h}\|^2_2 \tag{6}$$

$$\hat{\boldsymbol{\delta p}}_{t:t+H} = f^{\text{DYN}}_\phi(\mathbf{s}_t, \mathbf{a}_{t:t+H}).$$

We trained the perception model with three data sources from different robots ("Kobuki + Jackal + Human" in Fig. 4), one of which is a Clearpath Jackal robot. We then deployed this perception model on the Jackal robot. The robot's objective is to go towards a goal location while avoiding collisions, using the same reward function for planning as Eqn. 5.

We also evaluated our method on a Parrot Bebop drone, shown in Fig. 3, which was not included in the data collection process for the perception model. Note that this generalization to a new platform was only possible because the drone's image observations lies in-distribution within the perception training data. The drone's objective is to avoid collisions with minimal turning. The reward function for planning is

$$R(\hat{r}_{t:t+H}, \hat{\boldsymbol{\delta p}}_{t:t+H}, \mathbf{a}_{t:t+H}, \mathbf{g}_t) \tag{7}$$

$$= \sum_{h=0}^{H-1} \hat{r}_{t+h} - \|\hat{\boldsymbol{\delta p}}_{t+h}\|_1.$$

The results in Fig. 6 compare HInt against the conventional hierarchy approach on a Jackal robot with its normal dynamics and with its steering limited to 40% of its full steering range, as well as on a Bebop drone flying at low speed (8 degrees forward tilt) and high speed (16 degrees forward tilt). While the two approaches were able to achieve similar performance when evaluated on the normal Jackal and the low speed drone, our approach outperformed conventional hierarchy on the limited steering Jackal and on the high speed drone. The poor performance of the conventional hierarchy baseline in these experiments can be explained by the inability of the conventional higher-level planner – which plans kinematic paths based on visual observations – to account for the different dynamics limitations of each platform. This can be particularly problematic near obstacles, where the kinematic model might decide that a last-minute turn to avoid an obstacle may be feasible, when in fact the dynamics of the current robot make it impossible. This is not an issue for the standard Jackal robot and the low speed drone, which are both able to make sharp turns. However, the limited steering Jackal was unable to physically achieve the sharp turns commanded by the higher-level model in the baseline method. Similarly, due to the aerodynamics of high-speed flight, the Bebop drone was also unable to achieve the sharp turns commanded by the baseline higher-level planner, leading to collision (see, e.g., the lower-right plot in Fig. 6). In contrast, in HInt, because the low-level model is able to inform the high-level model about the physical capabilities of the robot, the integrated model could correctly deduce that avoiding the obstacles required starting the turn earlier when controlling a less maneuverable robot, which allowed for successful collision avoidance.

| | Normal Jackal | Limited Steering Jackal | Low Speed Drone | High Speed Drone |
|---|---|---|---|---|
| Conventional Hierarchy | **80%** | 0% | **100%** | 20% |
| HInt (ours) | **80%** | **100%** | **100%** | **100%** |

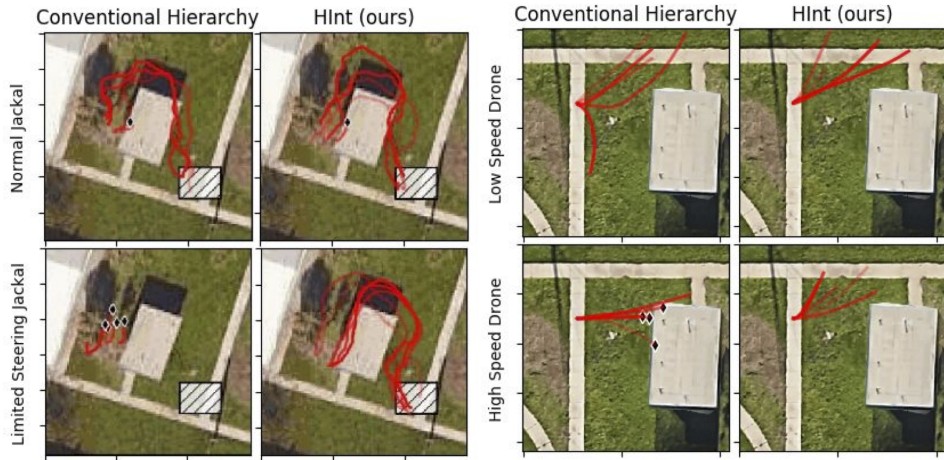

Figure 6: Comparison of conventional hierarchy vs. HInt (ours) approaches in a real world experiment, on Jackal with normal dynamics and with its steering limited to 40%, as well as on a drone with low speed (8 degrees forward tilt) and high speed (16 degrees forward tilt). Both approaches were evaluated with 5 trials each from the same starting position. Our approach is able to achieve higher success rates for reaching the goal region when deployed on the limited steering Jackal and the drone, because the perception model was able to reason about its dynamical limitations.

We also evaluated HInt and conventional hierarchy on a visual navigation task in simulation involving robot cars with more drastic dynamical differences, such as limited steering, right turn only, and 0.25 seconds lag. More details can be found in the Appendix (see §E). In both the real world and simulation, we showed that conventional hierarchy can fail when deployed on dynamical systems that are not a part of the perception dataset. This is because the higher level perception-based policy can set waypoints for the lower level dynamics-based policy that are outside the physical capabilities of the robot. In contrast, HInt's integrated planning approach enables the dynamics model to inform the perception model about which maneuvers are feasible.

## 5 Discussion

We presented HInt, a deep reinforcement learning algorithm with hierarchically integrated models. The hierarchical training procedure enables the perception model to be trained on heterogeneous datasets, which is crucial for learning image-based neural network control policies, while the integrated model at test time ensures the perception model only considers trajectories that are dynamically feasible. Our experiments show that HInt can outperform both single-source and conventional hierarchy methods on real-world navigation tasks.

One of the key algorithmic ideas of HInt is that sharing perception data from heterogeneous platforms could improve generalization and reduce the amount of training data needed from the deployment robot. As with all machine learning methods, our approach generalizes to situations that are within the distribution of the training data. So if training data comes from a variety of ground robots with similar cameras, we expect good generalization to other ground robots with similar cameras, but not, for example, to walking robots. We conjecture that this problem can be solved by learning from a bigger and more diverse dataset. An exciting direction for future work could be to scale up our experimental evaluation to a much larger number and diversity of platforms (e.g. training the perception model with data from 1000 different cars, drones, and legged robots). Such an approach could lead to a general and robust perception system that can be directly deployed on a wide range of mobile robots without needing to collect additional perception data from the deployment robot. We believe that enabling robot platforms to learn from large and diverse datasets is essential for the success of robot learning, and that HInt is a promising step towards this goal.

**Acknowledgments**

We thank Anusha Nagabandi and Simin Liu for insightful discussions, and anonymous reviewers at RAIL for feedback on early drafts on the paper. This research was supported by DARPA Assured Autonomy and ARL DCIST CRA W911NF-17-2-0181. KK and GK are supported by the NSF GRFP.

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
