# OpenReview forum: "Hierarchically Integrated Models: Learning to Navigate from Heterogeneous Robots"
_robot-learning.org/CoRL/2021/Conference — CoRL2021 Poster_

### Official Review · Reviewer_JJFZ · 2021-07-20

**Originality:** Good
**Technical Quality:** Good
**Clarity Of Presentation:** Very Good
**Impact:** 3

**Recommendation:**

Weak Accept: I recommend accepting the paper, but will not argue for my recommendation if the majority of other reviewers have a different opinion.

**Summary:**

This paper proposed a novel DRL framework to efficiently share multiple robots' samples. The proposed framework has two separate models: perception and dynamics model to learn the observation-reward map and the robot dynamics respectively. This double-model framework enable the agent to learn a wide range of observation while considering the current robot behaviors. The experiment was conducted in multiple environments and robots and clearly show the advantages of the proposed framework.

**Issues:**

All issues have been pointed in weaknesses.

**Reviewer Expertise:**

Very good: Comprehensive knowledge of the area

**Strengths And Weaknesses:**

Strengths
1. The paper clearly shows its motivation and contribution. It has a well-written introduction and related works.
2. The approach section is well designed with figures.
3. The experiment is attractive and informative. The support materials are helpful to understand the details of experiment.

Weaknesses
The reviewer's concern are following:
1. Sharing samples collected from multiple robots is a good idea. However, the reviewer wonders whether and how the image samples collected from multiple robots are suitable to guide a specific one in an unknown environment. From Fig. 3, the cameras were fixed in different position of different robots, therefore, the samples collected from some robots (drone) may never be seen by others (UAV). How did the experience of a drone turns to help a UAV in an area explored by the drone only? Although the results show a improvement, the generalization capability of perception model should be further addressed.
2. For the dynamics model, the reviewer found that the state s, one input of f^{DYN} was set to empty ("There is no inputted state"). If so, the reviewer believes f^{DYN} may only learned very limited "dynamics of each robot" which may be insufficient to predict robots' movement. The author should further explain and discuss it with details (why set state to empty).
3. The reviewer also find that the baseline approach had a poor performances when the sample collection and evaluation are both in Jackal (Urban). It should have a close performance to the proposed approach since there is no samples collected by other robot/environment. The author should give a reasonable explanation and make the comparison fair if necessary.

**Summary Of Recommendation:**

This paper has clear contribution. It can be accepted once the pointing issues are addressed.

---

> ### Author Response · Authors · 2021-08-25
> **Response to Reviewer JJFZ**
>
> We thank you for your comments and feedback. Here, we provide discussion on the generalization capabilities of the perception model, the accuracy of the dynamics models, and the performance of the single-source approach on the Jackal robot.
>
> Sharing perception data across different platforms was able to improve performance, because, in the same way that ImageNet pre-training is effective for solving downstream tasks for the computer vision community (https://arxiv.org/abs/1608.08614, https://arxiv.org/abs/1612.01452, https://openaccess.thecvf.com/content_ICCV_2019/html/He_Rethinking_ImageNet_Pre-Training_ICCV_2019_paper.html ), the representations extracted by learning from samples collected from one robot can be useful for another robot. Even if the camera images and deployment environment are not exactly the same for the two robots, there are still many underlying visual features of the world that are shared. The main goal of this work is to propose a method to share data between different robot platforms based on this property of perception models, rather than to rigorously analyze the generalization capabilities of computer vision systems, though this has been studied in prior work (https://arxiv.org/abs/1608.08614).
>
> Regarding state inputs to the dynamics model, we were able to accurately model the dynamics of the robots without additional state information for the platforms that we experimented on. This is evident from the success of our HInt policies in controlling the robots to complete the desired task. However, the dynamics of other platforms (that we did not experiment on) might depend on additional state information, for example the current battery level or velocity. Thus, we include the state s in our formulation of f^{DYN} to account for this more general setting.
>
> Our HInt approach was able to perform better than the single-source approach even when the data collection and evaluation were both on the Jackal robot, because HInt was trained on a larger and more diverse dataset, which helps the model generalize more broadly. The single-source Jackal (Urban) model was trained on 3.5 hours of data. In comparison, the HInt Kobuki (Office) + Jackal (Urban) model was trained on 7.2 hours of data, and the HInt Kobuki (Office) + Jackal (Urban) + Human (Industrial) model was trained on 8.4 hours of data, because HInt is able to leverage data collected from platforms different from the deployment platform. Even though the Kobuki and Human datasets were collected from different environments, they were also performing navigation and contained collision reward signals, which can help the perception system learn to make better collision predictions across all environments.

---

### Official Review · Reviewer_DdRP · 2021-07-24

**Originality:** Very Good
**Technical Quality:** Good
**Clarity Of Presentation:** Good
**Impact:** 4

**Recommendation:**

Weak Accept: I recommend accepting the paper, but will not argue for my recommendation if the majority of other reviewers have a different opinion.

**Summary:**

The authors propose a new machine learning paradigm for building autonomous robot navigation systems. Specifically, they propose HInt, a hierarchical approach which consists of two separately learned models: (1) a high-level "perception" module, and (2) a low-level "dynamics" module. The authors propose to train (1) from data gathered from potentially multiple robots/environments, and to train (2) using data gathered on the deployment platform itself. During deployment, the composition of these learned models is used to define a scoring function that a planner may use when searching for control sequences. The authors validate their method using real-world experimental data.

**Issues:**

I'm willing to revise my score upwards if the authors are able address the issues laid out in the "weaknesses" section above.

**Reviewer Expertise:**

Very good: Comprehensive knowledge of the area

**Strengths And Weaknesses:**

STRENGTHS

(S1) The authors have focused on an important problem in the robotics community (vision-based navigation), and have proposed a novel machine learning method with an important benefit (ability to leverage traning data from heterogenous platforms) to address it.

(S2) The authors have performed an impressive number of real-world experiments that have yielded data to support their claims.

WEAKNESSES

(W1) The presentation of the method itself is a bit unclear:

a. First, very little upfront intuition is given regarding why, exactly, one should expect the high-level perception module to "work" (ie, provide meaningful scores) at all. My guess is that the image and a short enough sequence of pose changes provides enough information to tell whether or not the robot will collide and/or make progress to the goal. If that's correct, the authors should very clearly state this upfront or, if not, the authors should provide some other intuition regarding why we could expect a function with these inputs to reasonably be able to learn what they claim it learns.

b. Second, the presentation of the concepts surrounding Equation (1) is a bit murky. It's not until much later in the paper that one learns where the (supervised) reward information comes from, and that seems very important to the method.

(W2) The experiments section is missing important information regarding how, exactly, the training data is collected. This is somewhat related with (W1b) above--if the method uses supervised learning for the reward prediction problem, then it stands to reason that a variety of rewards _must_ be present in the dataset, including a potentially large number of collisions. Can the authors comment on this? Regardless of the answer, more information regarding how training data was/should be collected needs to be provided to the reader.

(W3) I find the title to be a bit misleading in the following sense. Typically, the phrase "multi-robot," like "multi-agent," refers to methods that seek to address problems that involve the simultaneous control of multiple agents. Here, instead, it appears that the authors are using the term to indicate that many robots are used to _gather training data_. While the proposed method is indeed an interesting contribution to the literature, I think the term "multi-robot deep reinforcement learning" should be reserved for methods that control multiple agents, as seems to be standard in the community. The authors should revise the title--perhaps the term "heterogeneous" could be more appropriate here?

(W4) While I applaud the authors for the impressive real-world evaluation done here, I do wonder how the results should be interpreted in the context of autonomous navigation more broadly. What does the proposed system (e.g., a vision-based navigation system) allow the system to do in these environments that could not be accomplished by existing robot navigation systems from across the entire community? For example, is there something challenging about these environments where the requested navigation tasks could not be accomplished by a classical planning-based system using lidar? I think the authors should more clearly articulate--either in the experiments section or perhaps in the introduction--the exact benefit that machine learning provides here.

MINOR COMMENTS

(MC1) Figure 4 is missing important information. In particular, it's not clear what exactly is going on in the part of the figure with the aerial photos. What does each photograph correspond to? The authors have stated in the caption that red/blue/green indicate trials starting from different positions, but which environment does each row correspond to? Does the word above each picture correspond to the training data used for the trials shown? I couldn't find answers to these questions in either the caption or the text.

(MC2) The "angle" symbol in Equation (4) needs to be precisely defined.

POST-RESPONSE COMMENTS:

Thanks to the authors for providing additional detail and addressing many of my concerns. I'm willing to raise my score conditioned on adding detail (either in the main paper or the appendix) about just how many collisions were required in the training data in order to achieve the results shown in the paper. I think this is a critical detail.

**Summary Of Recommendation:**

My current recommendation is "weak reject." I very much like the overall thrust of the work, and I applaud the authors for proposing an interesting new method that can potentially allow system designers to take advantage of a wide variety of vehicle video data in service of learning for navigation. That said, I think there are several important ways in which the paper needs to be improved before I would feel comfortable arguing for acceptance.

---

> ### Author Response · Authors · 2021-08-25
> **Response to Reviewer DdRP**
>
> We thank you for your comments and feedback. Here, we provide discussion on how the perception model makes accurate predictions, and details on how the data was collected and labeled.  We also provide a clarification on the scope of our work concerning the role of learning in autonomous navigation. We believe that these responses largely resolve the issues raised in your review, but if they do not, we would appreciate further feedback.
>
> With regards to the perception model (W1a), the current image combined with future positions of the vehicle is sufficient to predict whether a collision will occur in most cases, except when those future positions pass outside the field of view or behind occlusions. You are right that this is generally the case for relatively short sequences (in our case, we plan over horizons of 2.5 seconds). Note that prior work has already confirmed that such an approach (in the single robot setting) can work well for collision avoidance [34]. The idea of using such an approach for collision avoidance (and any accompanying limitation) is not new in our paper, our contribution is to extend this to learn from multiple distinct robots. We’ve updated the paper to clarify that, while this approach does result in a limitation for long-horizon navigation, our aim is to perform short-horizon obstacle avoidance.
>
> With regards to reward generation (W1b), labels for the collision reward were collected using onboard collision sensors or manual labeling (Table S4). We have updated our paper to include this information when explaining equation 1.
>
> With regards to the training data (W2), we collected real-world perception and dynamics data by running the robot with a correlated random walk policy in their respective environments (analogously to the approach in [34]). Our dataset contains 560 collisions/ 50,000 timesteps for the Jackal, 2267 collisions / 32,884 timesteps for the Kobuki, and 204 collisions / 12,523 timesteps for the person with the GoPro camera. Since the dataset is highly imbalanced, we rebalance each batch during training, such that 50% of the trajectories contain a collision label and 50% of trajectories do not, in order to ensure that there would be enough diversity in rewards in the data during training (see Table S5). While it's true that the perception model needs labels of collisions to learn about collisions (and, analogously, labels for whatever other reward signal it might be asked to predict), we believe that our approach actually alleviates this burden as compared to prior work, such as [34], since the perception system can be trained using data from other robots (e.g., we could use data with collisions from a ground robot, which is comparatively safer, and use it with a drone). However, we will make the requirement for collisions in the data more explicit in the paper, though we note this limitation is similar to prior work [34, https://arxiv.org/abs/1902.03701, https://arxiv.org/abs/1709.10489].
>
> For other details, please see Appendix A for details about the structure of the training data (dimensionality of inputs, outputs, etc.), and Appendix B.1 and C.1 for details about the data collection process, including the environment, time discretization, data collection policy, size of dataset, and reward and pose label generation procedure. If there are other specific details that the reviewer would like to see, we would be happy to provide them. Unfortunately, due to the very restrictive CoRL page limit, many of these details will need to go in the appendix, but we agree that it’s important to provide all the details necessary to make the work reproducible, and if there is anything else that you believe is missing, please let us know and we will add it!
>
> Regarding our use of a vision-based navigation system (W4), we acknowledge that the question of whether robotic navigation should use learning methods or not is a complex one. We are not attempting to answer that question -- we do not claim that our method is a better way to solve navigation problems than non-learning methods, only that our approach is a better way to integrate data from multiple robots for systems that aim to use learning for navigation. There is a large literature on learning to navigate from images  [34, https://arxiv.org/abs/2010.10903, https://arxiv.org/abs/1807.05211, http://rpg.ifi.uzh.ch/docs/RAL18_Loquercio.pdf, https://arxiv.org/abs/1903.02749, https://arxiv.org/abs/1709.10489, https://arxiv.org/abs/1903.02531, https://arxiv.org/abs/1902.03701], and therefore we think that better learning-enabled vision-based navigation methods are of interest to the community.
>
> Finally, we have updated our paper to include a more clear description of Figure 4 (MC1), and the angle symbol in Equation 4 (MC2). We’ve also updated the title to be “Hierarchically Integrated Models: Learning to Navigate from Heterogeneous Robots” to address your concern about the term “multi-robot” (W3).

---

> > ### Comment · Reviewer_DdRP · 2021-09-03
> > **Thanks**
> >
> > Thanks to the authors for the thorough response. I believe most of my concerns were addressed here by providing more detail here in the discussion, though I reviewed the revised paper and supplemental material and still could not find any mention of how many collisions were required in the dataset there for a potential reader. I believe this information is important to readers, and would like to see it added into the final revision of the paper.
> >
> > Assuming that will happen, I'm happy to raise my score.

---

> > > ### Author Response · Authors · 2021-09-03
> > > **Response**
> > >
> > > Thank you for your response! We have added the following modification to the text in Section B.1 L392:395 to explicitly clarify how many collisions were available in the dataset.
> > >
> > > 	For our real world experiments (§4.1), perception datasets were collected by a Yujin Kobuki robot in  an office environment (2267 collisions), a Clearpath Jackal in an urban environment (560 collisions), and a person recording video with a GoPro camera in an industrial environment using correlated   random walk control policies (204 collisions).
> > >
> > > Although CoRL doesn’t allow us to upload a revision at this time, we will update the final version of the manuscript with this text.

---

### Official Review · Reviewer_KezA · 2021-07-25

**Originality:** Very Good
**Technical Quality:** Good
**Clarity Of Presentation:** Very Good
**Impact:** 4

**Recommendation:**

Weak Accept: I recommend accepting the paper, but will not argue for my recommendation if the majority of other reviewers have a different opinion.

**Summary:**

The authors address the problem of autonomous navigation on heterogeneous robotic platforms. One of the limitations of many current approaches is the need to train an algorithm in the same specific platform in which it is intended to be used. To overcome this problem, HInt, a new model-based Deep Reinforcement Learning approach, is proposed. HInt has two main components: the dynamics model, trained with data from a specific robotic platform, and the perception model, trained instead with data collected using different sources. Combining together these two components, the resulting model is able to take into account the dynamics of the robots in which it is deployed while, at the same time, leverage heterogeneous dataset gathered by different platforms.

**Issues:**

1) To help the reader to better understand the training procedure, a brief description of the stochastic zeroth-order optimization method is suggested;
2) Section 3.4, containing the algorithm summary, is useful as a recap, but should be accompanied by the algorithm pseudocode for better clarity;
3) Three versions of HInt trained with a single source dataset (i.e., Urban, Office, and Industrial) could be compared against its final version and [34].

**Reviewer Expertise:**

Very good: Comprehensive knowledge of the area

**Strengths And Weaknesses:**

The paper is in general well written and easy to follow. The problems addressed by the work are relevant to the Robotics community and are clearly defined by the authors. The literature review is also adequate. The proposed approach is interesting and promising even if requires a real-world data collection, which can be time and resource expensive. The training procedure and model structure are also clearly described but it would be useful to better explain the optimization method used to learn the reward signal and to add a more formal definition of the overall algorithm. Finally, the experiments show the performances of the proposed approach against a recent state-of-the-art method [34] in different real scenarios. While the comparison with the other method demonstrates the superior adaptation capabilities of HInt, it would be interesting to test it against a version of itself trained with a limited set data. In this way, the importance of a heterogeneous dataset for the perception model training would be explicitly measured.

[34] G. Kahn, P. Abbeel, and S. Levine. BADGR: An autonomous self-supervised learning-based navigation system. arXiv:2002.05700, 2020.

**Summary Of Recommendation:**

The paper is original, technically sound and clearly written. The proposed approach is interesting and my overall recommendation is positive.

---

> ### Author Response · Authors · 2021-08-25
> **Response to Reviewer KezA**
>
> Thank you for your comments and feedback.
>
> First, we would like to provide a clarification on our experiment which compares HInt against the single-source approach. In our experiment, we used the exact same training architecture and optimization parameters for the two methods, with the only difference being the training data used. Thus, the single-source approach is a version of HInt trained with a limited set of data. The original submission was unfortunately unclear on this point. We have updated our paper to more clearly explain the single-source approach.
>
> Regarding comparisons against the single-source approach in additional environments, our current experiments compare HInt against the single source approach trained on data from the evaluation environment, and on data from the office environment (for fair comparison, since we did not evaluate on the office environment). We believe that the additional single-source experiments (i.e. learning from industrial data and deploying on urban, and learning from urban data and deploying on industrial) are unlikely to perform better than the existing single-source experiments (i.e.learning from industrial data and deploying on industrial, and learning from urban data and deploying on urban). Running each real-world experiment requires significant time and labor (each perception model requires ~5 hours to train, and each starting location requires ~3 hours to evaluate due to time needed for setup, resets, and battery charge), and we believe that the insights gleaned from running them may not be worth the amount of work.
>
> We’ve updated our paper to include a description of the stochastic zeroth-order optimization used in the methods section.

---

### Official Review · Reviewer_cpmD · 2021-07-25

**Originality:** Fair
**Technical Quality:** Fair
**Clarity Of Presentation:** Good
**Impact:** 2

**Recommendation:**

Weak Reject: I recommend rejecting the paper, but will not argue for my recommendation if the majority of other reviewers have a different opinion.

**Summary:**

This paper presents a control strategy for mobile robots that leverages a neural network forward model in an MPC fashion. The model consists of two parts: a perception module which is trained using datasets collected from different robots and controllers and a dynamics model of a specific robot being controlled.
This paper tries to enrich the dataset for off-line/off-policy reinforcement learning for visual control tasks. Typically, when training a policy for a new robot, we have to create a new dataset collected by the same robot and low-level tracking controller. This paper tries to reduce this time-consuming process by reusing datasets collected by different platforms

The main contribution of this paper is the construction of modular control architecture that separates perception and motion prediction. The perception model predicts future rewards (collision event) given the future pose sequences. By construction, the perception model is independent of the dynamics of the robot & tracking controller and only focuses on the scene understanding. The motion prediction model (dynamics model) is trained separately for the tested robot to predict the future pose sequences given the future actions. The dynamics model learns the tracking behavior of the robot and its low-level controller. In the test time, the output of the dynamics model is fed to the perception model and a zeroth-order optimizer is used to generate action sequences that minimize the future rewards.

With experimental results, authors claim that:
    - (Q1) Using diverse datasets collected in different environments and using different robots improves the performance of a navigation policy.
    - (Q2) Proposed architecture performs better than the conventional hierarchical approaches.

**Issues:**

- The result in Figure 4 cannot be used to answer question Q1. The experiment in Q1 is not fairly comparing the Multi-source v.s. Single-source. The multi-source model and the single-source model that the paper is comparing have different control architectures. One is using a hierarchical model while the other is using an end-to-end model. For a fairer comparison, the authors have to compare two models of the same control architecture that are trained with multi-source data and single-source data. The effects of different architecture and different data sources are blended in the presented experiment.
- The dynamics model does not consider perception and can have a large variance. To predict future poses, we need environmental information such as ground friction or obstacle location. For example, when commanded to go straight into a wall, the future pose will differ depending on the distance to the wall. This sort of information cannot be predicted using the proposed dynamics model. One of the main claims of this paper is the decoupling of dynamics from the perception, which is supported by Q2, but the dynamics model should still be dependant on the perception. The paper should clearly present this limitation and in which environment/setup the datasets for dynamics models are collected.
- In the table in Figure 4, It would be better to clarify that "Jackal" is used for testing.
- Typo in like 242: "drone drone"

**Reviewer Expertise:**

Excellent: Expert knowledge on the topic of the paper

**Strengths And Weaknesses:**

- This paper is tackling an interesting problem. For autonomous driving systems, the problem is usually tackled by leveraging explicit intermediate representations such as semantic segmentation or lane detection. In this work, the pose sequence acts as an intermediate representation that connects the two separate models.
- The proposed method and experimental results seem to be limited to specific setups. All the datasets are collected from similar perspectives and the proposed approach cannot tackle one of the main problems of reusing existing data: different perspectives, FOVs, and image sensors across datasets. As shown by "Person in Industrial" in Fig. 3, the data used by this work seem to be collected in similar perspectives and the generalizability of the proposed approach is questionable.

**Summary Of Recommendation:**

The paper tackles an important problem in applying reinforcement learning using real-world data. The real-world experiments prove that the proposed architecture can improve the performance of a RL-trained policy as it enables using richer data and stabilizes the perceptive planning by only using (likely) feasible trajectories predicted by a dynamics model. However, the approach and experimental result cannot fully support the promises made in the introduction.

---

> ### Author Response · Authors · 2021-08-25
> **Response to Reviewer cpmD**
>
> Thank you for your comments and feedback. We believe that the main methodological concern about the work (in regard to Figure 4) stems from a subtle misunderstanding, which we clarify below. We also provide a discussion of the generalizability of our method, as well as the limitations of the dynamics model. We believe that these responses largely resolve the issues raised in your review, but if they do not, we would appreciate further feedback.
>
> Regarding the experimental comparison presented in Figure 4, the single-source results presented in the paper are in fact a direct comparison with HInt. We used the exact same training architecture and optimization parameters for the two methods, with the only difference being the training data used. This was unfortunately unclear in the submission, and we have updated our paper to more clearly explain the baseline method.
>
> Regarding the generalizability of our method, our work has an implicit assumption that the training data covers the space of perspectives that we hope to evaluate on. Thus, in order for the learned models to generalize to a new platform’s perception system, the dataset needs to include images with enough variation such that the new perception system would lie in-distribution in the training set. We made this assumption more clear in the paper. However, it’s worth noting that this limitation is shared by virtually all learning-enabled vision systems: learned models will generalize well to test points that come from the same distribution as training points, and generally will not generalize out of distribution. Our method does not aim to solve this. If there is broad variability in perspectives in training, it should generalize over perspectives, if there isn’t, then it won’t.
>
> Regarding the predictive power of the dynamics model, it is true that, in our work, the dynamics and perception systems are decoupled, and we updated the paper to discuss this. The perception system can reason about the effect of observed obstacles on future states, as demonstrated in our experiments on collision avoidance, but it does so in a robot-agnostic way. It is still possible to incorporate environmental information such as ground friction or obstacle location into our dynamics model, by incorporating this information as part of the state input into the low-level dynamics model. The only condition is that the state input needs to be sufficiently low-dimensional (while it is also possible to feed in entire images as the state into the low-level dynamics model, this will take away the data-efficiency benefits offered by our method). But in some sense this issue is fundamental for any method: if we have only a small amount of data for a particular robot, and its dynamics depends on images in a unique way, it is likely impossible to learn without either prior knowledge (i.e., low-dimensional information) or more data. This condition does put restrictions on the kind of tasks that can be used with our method. However, in many robotic tasks, the dynamics of the system can be accurately modeled using low-dimensional state information only (for example navigation). We updated our paper to make this condition more clear. However, we believe the method is still valuable despite this limitation, as illustrated by the experiments in the paper.

---

> > ### Author Response · Authors · 2021-09-03
> > **Response**
> >
> > We wanted to follow up and ask if our responses have helped address your concerns about our experimental comparison, the generalizability of our method and the predictive power for the learned dynamics model. We would be happy to clarify any remaining concerns.

---

### Author Response · Authors · 2021-08-30
**Follow up**

We wanted to follow up and ask if our responses have helped address the concerns that the reviewers have raised. We would be grateful if you can let us know if there are remaining concerns that we can clarify.

---

### Meta-Review · Area_Chair_MuMT · 2021-08-13

**Recommendation:** Accept (Poster)
**Confidence:** 4

**Metareview:**

This paper has quite mixed reviews. On the one hand, the experiments on real robots show the value of a decoupled dynamics/collision architecture in generalization. However, the reviewers raised concerns about generality, how exactly the model is trained, and more broadly and fundamentally, about why this is necessary when quite conventional approaches have been doing similar things in robotics for decades. Addressing these questions in the rebuttal would be very helpful for the subsequent discussion.

--- post-rebuttal, some of the reviewers have raised their scores, and we are on a majority accept. Having looked at the authors' rebuttal, I believe they have answered most of the outstanding questions raised.

---

> ### Author Response · Authors · 2021-08-25
> **Response to Meta Review**
>
> Thank you for the summary of reviewer feedback.
>
> Regarding details about model training, we have updated the paper to include specific suggestions from the reviewers. We believe this generally addresses these concerns (insofar as is possible in an 8-page paper), but if there are any other concerns regarding the method, we would be happy to clarify or provide additional details.
>
> Regarding learning vs. conventional approaches, we are not claiming that our method is a better approach across the board to solve autonomous navigation compared to non-learning methods. Proving that our approach is better than any non-learning-based approach seems like comparatively a much higher bar, and we are not claiming that. But given that CoRL focuses on robotic learning, and there are many papers on learning-enabled navigation (that don’t compare to non-learning methods), we believe this scoping is reasonable. The goal of our work is to instead propose a better way to incorporate perception training data from different platforms, which could lead to better generalization in learning-enabled vision-based navigation systems. There are a large number of prior works that investigate the use of learning in navigation from images [34, https://arxiv.org/abs/2010.10903, https://arxiv.org/abs/1807.05211, http://rpg.ifi.uzh.ch/docs/RAL18_Loquercio.pdf, https://arxiv.org/abs/1903.02749, https://arxiv.org/abs/1709.10489, https://arxiv.org/abs/1903.02531, https://arxiv.org/abs/1902.03701], with benefits compared to conventional approaches such as fewer requirements on hardware sensors and better performance on rugged terrain. Such learning methods are often bottlenecked by availability of data, and this bottleneck could be alleviated by combining data from different robots. Thus, we believe that our work, which enables better learned-based navigation, will be of interest to the community.
>
> Regarding the generality of our method, we acknowledge that our method cannot generalize to new platforms for which the camera sensor has drastically different attributes from those in the training data distribution, such as perspective, POV, etc. However, handling distributional shift is a distinct issue from the one we tackle -- as with most learning methods, the goal is to generalize within distribution (i.e., to new robots with sensors that are similar to those in training). We have updated the paper to make this limitation of our method more clear. We would like to emphasize that this limitation is not unique to our method, but rather appears in any learning-based computer vision system -- when evaluated on new images with out-of-distribution perspectives, no learned model is guaranteed to generalize.

---

### Decision · Program_Chairs · 2021-09-13

**Decision:**

Accept (Poster)

**Comment:**

This paper has quite mixed reviews. On the one hand, the experiments on real robots show the value of a decoupled dynamics/collision architecture in generalization. However, the reviewers raised concerns about generality, how exactly the model is trained, and more broadly and fundamentally, about why this is necessary when quite conventional approaches have been doing similar things in robotics for decades. Addressing these questions in the rebuttal would be very helpful for the subsequent discussion.

--- post-rebuttal, some of the reviewers have raised their scores, and we are on a majority accept. Having looked at the authors' rebuttal, I believe they have answered most of the outstanding questions raised.